# Terpenoid Biosynthesis Dominates among Secondary Metabolite Clusters in *Mucoromycotina* Genomes

**DOI:** 10.3390/jof7040285

**Published:** 2021-04-09

**Authors:** Grzegorz Koczyk, Julia Pawłowska, Anna Muszewska

**Affiliations:** 1Department of Biometry and Bioinformatics, Institute of Plant Genetics, Polish Academy of Sciences, Strzeszynska 34, 60-479 Poznan, Poland; 2Institute of Evolutionary Biology, Faculty of Biology, Biological and Chemical Research Centre, University of Warsaw, Zwirki i Wigury 101, 02-089 Warsaw, Poland; julia.z.pawlowska@uw.edu.pl; 3Institute of Biochemistry and Biophysics, Polish Academy of Sciences, Pawinskiego 5A, 02-106 Warsaw, Poland

**Keywords:** *Mucoromycotina*, secondary metabolite, natural products, SM biosynthesis, terpenes, NRPS, adenylate forming reductases, malpicyclins, malpibaldins

## Abstract

Early-diverging fungi harbour unprecedented diversity in terms of living forms, biological traits and genome architecture. Before the sequencing era, non-Dikarya fungi were considered unable to produce secondary metabolites (SM); however, this perspective is changing. The main classes of secondary metabolites in fungi include polyketides, nonribosomal peptides, terpenoids and siderophores that serve different biological roles, including iron chelation and plant growth promotion. The same classes of SM are reported for representatives of early-diverging fungal lineages. Encouraged by the advancement in the field, we carried out a systematic survey of SM in *Mucoromycotina* and corroborated the presence of various SM clusters (SMCs) within the phylum. Among the core findings, considerable representation of terpene and nonribosomal peptide synthetase (NRPS)-like candidate SMCs was found. Terpene clusters with diverse domain composition and potentially highly variable products dominated the landscape of candidate SMCs. A uniform low-copy distribution of siderophore clusters was observed among most assemblies. *Mortierellomycotina* are highlighted as the most potent SMC producers among the *Mucoromycota* and as a source of novel peptide products. SMC identification is dependent on gene model quality and can be successfully performed on a batch scale with genomes of different quality and completeness.

## 1. Introduction

Early-diverging fungi were long considered to be devoid of secondary metabolite (SM) clusters; however, this perspective has gradually changed with the advancement of genome sequencing of selected representatives. The reviews by Kerstin Voigt, Ekaterina Shelest and colleagues [1,2] showed that sequences similar to known secondary metabolite clusters (SMCs) can be identified in most analysed *Mucoromycota* genomes. The identified SMCs harboured key enzymes involved in SM biosynthesis in *Dikarya*: polyketide synthases (PKS), nonribosomal peptide synthetases (NRPS), terpene cyclases/synthases (TC), and dimethylallyl tryptophan synthases (DMATS). One or several of the core genes are typically surrounded by accessory genes, most often encoding additional enzymes (e.g., transferases, hydroxylases), regulatory transcriptional factors and/or efflux transporters.

The main classes of secondary metabolites in fungi include the respective compounds tied to the above core enzymes: polyketides, nonribosomal peptides, terpenoids and siderophores. The polyketides are produced by PKS clusters coding several proteins forming a multi-domain enzyme that condensates acyl-CoA precursors. Polyketides are typically modified by accessory enzymes, leading to the formation of complex structures. Aflatoxin, patulin and lovastatin belong to the best-studied fungal polyketides [3]. Nonribosomal peptides are catalysed without a mRNA template by a complex enzyme called NRPS. The peptide often includes noncanonical amino acids and is modified by accessory enzymes similar to polyketides. Well-known NRPS include β-lactam antibiotics, cyclosporine A and echinocandin [4]. Terpenes are isoprene polymers which often have multicyclic structures and can also be further modified, leading to the formation of terpenoids. Terpenes and terpenoids often possess scent, flavour or colour, like carotenes. Terpenoid compounds play diverse functional roles: they can serve as toxins (e.g., fusicoccin A of *Diaporthe amygdali*) or growth-promoting factors (e.g., gibberellin GA14 of *Fusarium fujikuroi*) [5]. Siderophores are produced for the iron chelation needed for both storage and transport of this element. Iron is usually scarce, and many microorganisms produce siderophores to scavenge insoluble iron forms, e.g., fusarinines of *Fusarium* spp., coprogen of *Pilobolus* sp. and asperchromes B1-F3 of *Aspergillus ochraceus* [6]. Although siderophores are not a biosynthetic class of SMs on its own, they are biosynthesized by an NRPS in the case of hydroxamate-based siderophores (preferably in ascomycetes (e.g., triacetylfusarinine C) and basidiomycetes (e.g., basidioferrin)) or by NRPS-independent siderophore (NIS) biosynthetic enzymes in the case of polycarboxylate siderophores (preferably in early-diverging fungi such as *Mucoromycota*).

For example, siderophore production has been confirmed in *Glomus* (glomuferrin) [7], *Apophysomyces* [8] and *Rhizopus delmar* (rhizoferrin) [9]. The latter has been further characterised. It belongs to the NIS family [9] and is likely to be regulated by glycosylation [10] (see the example in Figure 1). Experimental studies showed that different *Mortierellomycotina* representatives produce diverse natural products with interesting properties e.g., malpinin acetylated hexapeptides and malpibaldin cyclic pentapeptides [11]. These compounds have been linked to the presence of several NRPS clusters, for example, in the mycelium of *Mortierella alpina*. The NRPS of *M. alpina* are related to bacterial ones and contain atypical epimerase/condensation domains [12]. *Podila minutissima* (syn. *Mortierella minutissima*) is used for an enantioselective bioconversion of limonene to perilla alcohol and perillaldehyde [13]. PKS and NRPS cluster presence was recently described in transcriptomes of five *Mucor* species [14] and *Gigaspora margarita* [15]. The mycelium of *Mucor circinelloides* was shown to produce carotenoids and terpenoids [1]. This property of *M. circinelloides* is used in industrial settings. Finally, SMCs responsible for terpene biosynthesis were recently identified in several *Mucor* and *Rhizopus* genomes [9,16]. For instance, *Mucor irregularis* (syn. *Rhizomucor variabilis*) produces indole-diterpenes with a fused diterpene ring called rhizovarins [17].

The best-studied compound isolated from basal fungi is a macrolide, rhizoxin, produced by an endosymbiotic bacterium (*Mycetohabitans rhizoxinica)* of *Rhizopus microsporus* and encoded by a huge NRPS–PKS cluster [18]. Some fungi harbour the necessary genes themselves; others rely on their endohyphal bacteria (EHB). Similarly, in the *Linnemannia elongata* (syn. *Mortierella elongata)* and *Mycoavidus cysteinexigens* symbiosis, several compounds are produced, and the genome of the bacteria encodes NRPS and other metabolite clusters [19].

Fungal secondary metabolites can also promote plant growth in harsh conditions. This process is mediated by the release of phytohormones; for example, gibberellic acid (a diterpenoid acid) produced by *Rhizopus stolonifer* [20]), *Mortierella antarctica* and *Podila verticillata* (syn. *Mortierella verticillata*) [21]. Moreover, the plant–fungus interaction can lead to the synthesis of novel compounds which can be produced neither by the host nor the endophyte alone and can vary depending on the host plant species, which have been shown in, for example, a *Umbelopsis dimorpha* growing on *Kadsura angustifolia* and wheat bran [22].

Encouraged by the relative scarcity of data on SMCs for fungi other than *Dikarya* and the abundance of available genomic sequences, we aimed to perform a systematic screening of 157 publicly available *Mucoromycota* assemblies for the presence of secondary metabolite biosynthetic gene clusters. We applied a scalable procedure using publicly available tools for a batch evaluation of the diversity of SMCs within this group of fungi.

## 2. Materials and Methods

### 2.1. Candidate Secondary Metabolite Cluster Prediction

*Mucoromycota* genomes were downloaded from THE NCBI FTP on 8 June 2020 [23] (all assemblies with their references are listed in Appendix A). Subsequently, AntiSmash 5.1.2 [24] was used to annotate putative secondary metabolite clusters (SMCs), based on the built-in GLIMMERHMM [25] prediction of gene models (additional command line parameters: “--asf --cb-general --cb-subclusters --cb-knownclusters --smcog-trees --cf-create-clusters --fullhmmer --pfam2go --taxon fungi --genefinding-tool glimmerhmm”). Where possible, an additional round of AntiSmash 5.1.2 was run using existing GFF-formatted annotation. As indicated above, ClusterFinder extension was conducted to extend SMC boundaries. SMC type labels (e.g., ‘NRPS’, ‘NRPS-like’) were assigned based on AntiSmash’S own heuristics (strict and relaxed rule sets). To ensure specificity and sensitivity, protein sequences were scanned with rps-blast against the CDD [26] and PFAM [27] databases and filtered with a list of protein domains used by ClusterFinder rules in additional supervised checks that consisted of manual inspection of domain composition, leading to renaming of the Markov clusters (Appendix A)

All predicted SMCs were separated into individual GenBank files (Appendix A lists all the genomic coordinates of the SMCs) and CDS sequences were extracted with the gbf2xml and xtract utilities.

### 2.2. SMC-Encoded Protein Clustering and Annotation

All putative SMC-encoded proteins were clustered with MCL version 14-137 [28] in a process described in [29]. Based on a sampling of Markov Cluster Algorithm (MCL) cluster silhouette width, a threshold E-value of 1 × 10^−10^ and an inflation parameter of I = 1.55 were chosen for the final clustering. In parallel, all proteins encoded in each MCL cluster were compared against UniProt/SwissProt (release 2020_03) and MIBIG proteins (release 2.0) with DIAMOND v0.9.34.135 [30]. Hits were computed requiring over 50% coverage of both query and hit protein, using the sensitive mode of DIAMOND. Protein domains and conserved sites were annotated with InterPro (InterProScan-5.45.80 with additional models from PANTHER version 14.1; matches against all optional databases, as well as predictions of coiled coils were conducted. MCL clusters containing representative proteins from at least 10 candidate SMCs were further investigated and labelled manually based on protein domain annotation. Additionally, smaller MCL clusters (at least 2 proteins) were annotated if any DIAMOND hits to the MIBIG ‘biosynthetic’, ‘biosynthetic-additional’ or ‘transporter’ gene categories were found. The annotation results can be found in the Appendix A, manual annotation of candidate families; Appendix A, tally of all found similarities to domains/proteins).

All representatives of candidate families representing nonribosomal peptide synthases (NRPS), putative adenylate-forming reductases (called also non-(nonribosomal)peptide synthase-NPS, i.e., NPS_1, NPS_2, LYS2), luciferases (LUC) and PKS/NRPS hybrids (PKS–NRPS) were further annotated in AdenylPred [31]. Notably, this prediction was initially only possible for 380 sequences (out of 562 in total) where the AMP-binding domain matched the underlying HMMer model with sufficient quality. Upon relaxing the HMMSEARCH bitscore threshold (set at 50 by default) and separating out individual adenylate-forming domains, further predictions were obtained. All individual predictions, regardless of the HMMer score, are listed in the Appendix A (description of SMCs) and Appendix A. We performed this prediction for all proteins clustered in the following candidate families harbouring InterPro matches to adenylate-forming domains: MCL0001-NPS, MCL0003-LYS2, MCL0006-NRPS, MCL0048-PKS/NRPS and MCL0862-NPS_2. Malpicyclin/malpibaldin candidate SMCs were inspected based on BLASTP matches to the reference sequences (QOW41315.1, malpicyclin synthetase; QOW41314.1, malpibaldin synthetase) [32].

### 2.3. SMC Similarity Network Construction and Visualization

A similarity network was constructed based on an all vs. all SMC candidate comparison. A modified MultiGeneBlast 1.13 [33] *multigeneblast.py* script was used (available from https://github.com/gkoczyk/multigeneblast_gk, accessed on 15 December 2020), with minor fixes for the following: calculation of query coverage normalised to the length of the query sequence, and the possibility of calculating scores for single matches (by default, at least 2 syntenic hits were required for the detection of SMC similarity). A substantial number of SMCs were found to be either fragmented or incomplete. In particular, the nonpeptide siderophore biosynthetic regions frequently consist of only a single IucC/IucA homologue (IucA_IucC, PF04183). Thus, to properly capture the intercluster similarities, single matches were necessary for network construction. A list of thresholds was set, including a minimum percentage identity for hits of 30%, a minimum sequence coverage of 60% and a maximum distance between syntenic region boundaries of 1 Mbp.

The similarity network was visualised in CytoScape 3.7.2 [34]. The graph edges were weighted based on average sequence identity over all cluster sequences. The Prefuse Force Directed OpenCL layout was used, with 200 iterations plus 50 edge-repulsive iterations, as well as a spring coefficient of 5 × 10^−4^, the spring length set to 20 and the node mass set to 50. Deterministic layout starting from scratch was also enabled to ensure the consistency of future updates.

Phylogenetic tree reconstruction was performed with FastTree [35], sequence alignments were computed with mafft version 7 [36] and ambiguous columns were removed using Noisy [32]. Trees were visualised in iTol [37].

### 2.4. Detection of Genes of Putative Bacterial Origin

In the light of evidence of both endosymbiosis and horizontal gene transfer, we endeavoured to perform a background check of clusters/genes of likely bacterial origin. We utilised the Alien Index framework for this purpose [38]. Briefly, we first conducted a search of every protein-coding gene encoded in candidate SMCs vs. the nonredundant protein database (as of 19 March 2021) with DIAMOND. Default parameters were used for the search, except for taxonomic restriction, where only higher fungi (*Dikarya*) and bacterial hits were considered, and the minimum E-value threshold, which was set to 1e-05. Next, the average value of AI (Alien Index) was calculated for both the entire cluster and just its component genes from the core families (the fatty acid acyl-CoA AMP (FAAL) family was included among the core families for this test). These values are included in the Appendix A to this paper as, respectively, avg_AI and avg_AI_core (Appendix A).

## 3. Results and Discussion

### 3.1. Initial Screening

A grand total of 1605 SMCs were initially detected by combining de novo gene prediction and the existing GFF annotation (where possible) with ClusterFinder SMC extension. After double-checking for the presence of characteristic biosynthetic domain signatures (see Methods), the list was curated into a final set of 1215 SMCs with an assigned type of biosynthetic signature (Appendix A). Notably, inclusion of the existing annotation had a pronounced impact on SMC detection in some taxa. In *Glomerales* and *Diversisporales*, successful detection of SMCs was possible only in the presence of pre-existing gene annotation, with the sole exception of a single terpene cluster in *Gigaspora margarita* BEG34. Likewise, two predicted NRPS in *Hesseltinella vesiculosa* NRRL 3301 were erroneously joined into a single region and were curated manually (referred to as GCA_002104935__14_gff and GCA_002104935__15_gff in Appendix A).

Unexpectedly, in some cases, the application of de novo gene prediction also resulted in the detection of novel candidate SMCs. However, such examples were far fewer in number (26 SMCs vs. 345 stemming from the available annotation). A summary of the detection performance over the available genomes is available in Appendix A.

### 3.2. Limits of de Novo Prediction in the Absence of Reference Annotation

A parallel study on early-diverging fungal genomes [39] demonstrated a similar overall abundance of identified SMCs but with minor differences resulting from the already described differences in AntiSmash 4.0 and 5.11 performance and GFF annotation usage [40]. We observed the best SMC completeness for clusters detected on the basis of detailed reference annotation of gene structures. This was particularly evident for *Glomerales* representatives, where the majority of SMCs were identified solely when GFF annotation was supplied. This is largely explained by the AntiSmash settings for the annotation of fungal data. These included only one de novo gene finder (GLIMMERHMM) with splice signal models of the basidiomycete *Cryptococcus*. Interestingly, gene models seemed to have less impact on SMC identification in *Mucorales*.

### 3.3. De Novo Predicted SMC Candidate Sequences Are Poorly Characterised by Synteny

The clustering of proteins encoded in candidate SMCs (see Appendix A for characterisation of the detected domains and similarity to the reference sequences) reveals little overlap with the toolkit present in the MIBIG resource containing curated and confirmed sequences of biosynthetic clusters. In some cases (e.g., NRPS sequences), the low levels of sequence similarity can be attributed to the bacterial source of horizontal transfer (cf. malpibaldin MpbA and malpicyclin MpcA synthases [12]). However, inspection of the combinations of the candidate gene families showed over 60% potential SMCs (760) to be seemingly devoid of candidate accessory genes (defined as similar to MIBIG sequences, represented in multiple SMCs). Possible explanations, assuming correct gene prediction, include single horizontal transfers of evolutionarily distant clusters into fungal hosts (including endosymbiosis) and/or weak selective pressure leading to disjoint biosynthetic loci. As it is, the identified SMCs seem to be distantly related to known *Dikarya* SMCs, and recent publications support the incidence of horizontal gene transfer (HGT) [39,41].

### 3.4. Taxonomic Distribution

Genes similar to known secondary metabolite clusters were identified in the majority of analysed (*n* = 148 out of 157) *Mucoromycota* genomes (see Appendix A, Figure 2). Genomes of *Mortierellomycotina* encode an average of 16 SMCs per genome; the fewest were found in *Lobosporangium transversale* NRRL 3116 (*n* = 5) and *Mortierellales* strain BCC40632 (*n* = 3). Interestingly *Glomeromycotina* genomes differed in the number of SMCs, with half of *Glomerales* isolated devoid of SMCs but the *Diversisporales* having more than 20 SMCs. However, all seven *Glomerales* isolates without detected SMCs lacked gene annotation. *Mucoromycotina* genomes constituted the majority of the analysed dataset and, as expected, displayed intermediate content of SMCs, ranging from 22 SMCs in *Rhizopus azygosporus* CBS 357.93 to no SMCs candidates in *Cunninghamella bertholletiae*. Taxa with multiple sequenced strains revealed strong intraspecies diversity; for example, *Rhizopus oryzae* strains had from 2 up to 11 SMC candidates. This could be either due to assembly quality issues because SMCs tend to accumulate in repeat-rich regions, which are often missing from assemblies, or to real genetic diversity within the species.

Detection of SMCs also depends on the gene calling method and evolutionary distance to the reference genome used for training the gene calling tool. As mentioned before, this was particularly pronounced for *Glomeromycotina* genomes, which had no SMCs identified when the built-in GlimmerHMM (fungal version) was used. However, several clusters belonging to different types were reported when a GFF annotation was supplied. In consequence, we used genomic annotations when available.

### 3.5. Cluster Types

The greatest diversity of SMCs types was identified in *Mucorales*, which is likely a result of organism diversity and the relative abundance of assemblies for this order [42]. Canonical NRPS clusters are present predominantly in *Mortierellomycotina* (*n* = 66), whereas NRPS-like clusters with a phosphopantetheine attachment site (PP-binding, PF00550) or male sterility protein (NAD_binding_4, PF07993) and adenylate-forming (AMP-binding, PF00501) domains were identified in most of the analysed *Mucormycotina* assemblies. The largest category of SMCs is terpene clusters, which have diverse domain composition and potentially highly variable products. A uniform low-copy distribution of siderophore clusters was observed among most assemblies except for *Umbelopsis* spp.

Canonical PKS or PKS/NRPS clusters were not encoded by the analysed genomes except for *Hesseltinella vesiculosa* NRRL 3301, *Bifiguratus adelaidae* AZ0501, *Gigaspora margarita* BEG34 and all isolates of *Syncephalastrum racemosum*. It has been recently noticed that several of the putative clusters described in *Mucorinae* are actually fatty acid synthase (FAS) Type I genes [16]. However, our results show very few clusters related to fatty acid synthesis, with only 10 4’-phosphopantetheinyl transferase superfamily (ACPS, PF01648) domain copies in total. They were marked as “other” by the classifier and rejected from the final list.

Single SMCs of bacteriocin, betalactone, fungal-RiPP and ectoine were identified. Few of these were classified as hybrid SMCs with NRPS cluster fragments due to the presence of adenylation and condensation domains.

### 3.6. Presence of Bacterial-Like SMCs

We investigated the presence of genes of likely bacterial origin using the Alien Index concept (see Section 2.4). Apart from the clearly bacterial metabolite types (bacteriocins/ectoine), our results also support evidence of bacterial origins for candidate clusters among most NRPS-, PKS- and hybrid PKS/NRPS-containing clusters.

Conversely, the candidate NRPS-like clusters appear to be of fungal origin, with the exception of FAAL-containing clusters (*Glomerales, Diversisporales*), where the similarity of the core gene to the mycobacterial template was noted. Likewise, the clusters implicated in lysine (LYS2, single outlier) and siderophore (SID) biosynthesis were highly similar to their counterparts in *Dikarya*. In the case of terpenoids, no evidence of bacterial origin was retained in the AI values, except for a weak signal in the case of TS2-containing clusters among *Diversisporales*.

Putative bacteriocin biosynthesis clusters contained a domain of unknown function (DUF692, PF05114) coding ORF and displayed high sequence similarity to *Stenotrophomonas maltophilia* and *Pseudomonas* sp. sequences. In consequence, their occurrence in fungal genomes is likely from a bacterial source, either as contaminations or associated bacteria. The ectoine cluster present in the *R. oryzae* genome is the most similar to sequences from *Achromobacter marplatensis*, which is a member of *Burkholderiales*. The bacteria from this order are known as endohyphal partners of *Mucoromycota* representatives [18]. Therefore, we suppose that this SMC is likely of EHB origin. However, it remains an open question whether the abundance of *Mortierelles* and *Diversisporales* SMCs is dependent on their symbiotic status with endohyphal bacteria. Some previously named and characterised SMCs have clear sequence similarity to bacterial sequences [12]; others have a patchy taxonomic distribution without a clear evolutionary history. Other early-diverging fungi, *Basidiobolales*, seem to have acquired diverse SMCs from co-occurring bacteria [39]. The genomes of *Gigaspora margarita* and its *Candidatus Glomeribacter gigasporarum* symbiont encode NRPS-PKS hybrid clusters with genes of clear bacterial origin [15]. Interestingly, the sequence signatures are different for the SMC of the host and the symbiont [15]. The authors postulate an HGT scenario from unrelated bacterial donors leading to the emergence of the hybrid SMC in the associated bacteria and host fungi. Moreover, similar SMCs are present in related fungal isolates, pointing at a vertical inheritance following the SMC’s formation.

The supporting evidence for the discussed results is included in the Appendix A (Alien Index values for every cluster). Additionally, we have included a graphical summary of AI values broken down by taxa and cluster/core gene family in form of Appendix A.

### 3.7. SMC Sequence Clustering

A network of SMC nodes, created on basis of the MultiGeneBlast all-against-all search, showed the relatedness of SMCs belonging to one type and the interconnectedness of SMCs without prior categorisation (Figure 3; for MultiGeneBlast all-against-all scores, see Appendix A). While only a few SMCs lacked clear similarity to any other SMC (*n* = 3), this similarity was contingent on only a single matching gene in a further 176 cases and thus would be undetectable with a default MultiGeneBlast run (this was particularly frequent among siderophore biosynthetic regions, where 59 out of 155 showed no other genes in the matching region). In general, the similarity of core genes was the main driver for cluster similarity (a tendency also corroborated by the results of candidate family analysis, where there were relatively few large gene families shared by multiple SMCs).

Based on AntiSmash labelling, the SMC similarity network (Figure 3A) was dominated by terpene (*n* = 503, including 10 hybrid clusters) and NRPS-like SMCs (*n* = 421; 16 hybrid clusters). There were at least five major SMCs groups visible in the network, corresponding to: siderophores, NRPS and NRPS-like, and two related to terpenes. All of them are formed mainly by *Mucorales* sequences. NRPS-like sequences formed two distant groupings connected to the network of terpene SMCs and a separated NRPS cluster formed by *Mortierellales* sequences. Siderophore sequences grouped together, forming a well-separated cluster. Bacteriocins formed a separate subgraph, whereas PKS, betalactone and fungal-RiPP SMCs were located in the centre of the network, interconnected with terpene and NRPS clusters. Notably, the identity of visible SMC groups was properly refined by parallel annotation of the candidate gene families (Figure 3B), where conservation of core genes such as squalene synthase homologues was shown to explain the visible groupings (Figure 3B).

Notably, in the unsupervised Markov Cluster Algorithm (MCL) (see Appendix A for mapping of candidate families to individual clusters) clustering of all proteins encoded by constituent SMC member genes, no unambiguous core genes were detected in 141 cases. While the clusters as a whole passed the rule-based check of signature activities (e.g., presence of condensation and adenylation domains), an inspection of individual gene annotations (see Appendix A) typically pointed to distantly related gene families. For example, in NRPS-like clusters, the fatty acyl-AMP ligases (MCL0047, FAAL) fulfilled the detection rule requirements for nine clusters in the absence of a canonical adenylate reductase signature. We decided to retain such cases among the results, as, according to [43], their involvement in the biosynthesis of a particular metabolite is a likely prediction (e.g., unique complex lipids).

The presence/absence of common SMC types for the analysed genomes is summarised in Figure 4. Notably, the only cases of missing even common SMCs is limited to genomes lacking any form of GFF annotation (five genomes of *Rhizopus irregularis*, 2-*Cunninghamella bertholletiae*, 1-*Glomus cerebriforme*, 1-*Oehlia diaphana*; a further raw assembly of *Cunninghamella elegans* B9769 contains only a single terpene-like cluster candidate).

### 3.8. Predominance of Terpene Biosynthesis in Non-Dikarya Genomes

Terpenes are synthesised from isoprenyl diphosphates by terpene synthases. In fungi, terpenoids are predominantly produced from mevalonate. Isopentenyl diphosphate (IPP) and dimethylallyl diphosphate (DMAPP) are condensed into products of diverse size by an isoprenyl diphosphate synthase (IDS) and further modified by terpene synthase (TPS) [44].

This condensation of two geranylgeranyl pyrophosphate (GGPP) units results in the formation of *cis*-phytoene, which is further isomerized to a *trans*-phytoene linear polyene structure that is the basic element for further formation of all carotenes (or carotenoids in general). The subsequent activity of cyclases leads to the formation of rings on one or two ends of this chain, and the introduction of conjugated double bonds leads to the formation of the characteristic carotenoid structures (tetraterpenoids). The presence of conjugated double bonds is responsible for providing the characteristic yellow, orange or reddish colours of these hydrocarbons [45]. Apart from plants and bacteria, carotenoids are often present in some fungal lineages. Neurosporaxanthin produced by an ascomycetous fungus (*Neurospora crassa*) or astaxanthin synthesised by the basidiomycete yeast *Xanthophyllomyces dendrorhous* are good examples [46,47]. However, β-carotene production seems to be the most widespread in the fungal kingdom and is especially consistent within *Mucoromycotina*. Its synthesis has been studied, among others, in *Phycomyces blakesleeanus, Blakeslea trispora* and *Mucor circinelloides* [47]. All these different carotenoids in fungi can act as protecting agents against oxidative stress [48] or play a photoprotective role [49]. In *Mucoromycotina*, they are additionally involved in sexual reproduction, as carotenoids are precursors of trisporic acid (C_18_), which are basic building blocks of sex pheromones in the sexual reproduction of *Mucorales* [50]. Therefore, the predominance of terpene biosynthesis in this group of fungi seems to manifest.

Tabima et al. (2020) [39] described the dominance of terpene cyclases unique for this group of fungi. Our results support the importance of terpene biosynthesis in *Mucoromycotina*. Particularly, the presence of squalene- and geranylogeranyl phosphate synthase-coding genes (SQS, SQS_NDUF6 and GGPS) was observed throughout the phylogenetic tree of *Mucoromycota*, except for the majority of *Rhizopus* genomes, pointing out the probable loss of this character in this lineage or missing predictions due to a lack of dedicated gene annotation. SQS indicates the canonical squalene synthase found in fungal SMCs, whereas SQS_NDUF6 has similarity to NDUFAF6 (NDUF6, mitochondrial complex assembly factor 6), which is known to contain a nonfunctional squalene/phytoene synthase domain. Interestingly, the terpene biosynthesis-related genes were detected mainly within plant-related *Diversisporales* and *Glomerales*, while carotenoid synthesis-related genes were observed mainly within saprotrophic *Mucorales* (Figure 4). Although carotenoids belong to the tetraterpenoids, they form a well-characterised subgroup that, in several aspects, may be delimited and discussed separately.

Indeed, the retention of a conserved genomic cluster can best be seen among terpenoid SMCs, where carotenoid biosynthesis likely selects for tight linkage of carB and carRA orthologues. The crucial enzymes responsible for the condensation (geranyl diphosphate synthase, GPPS; polyprenyl_synt, PF003480), phytoene and/or squalene synthase activity (SQS_PSY, PF00494), desaturation (phytoene desaturase, PDS; amino-oxidase, PF01593)) and cyclisation (lycopene cyclase activity of LCPS; no Pfam, IPR017825, TIGR03462) steps are all accounted for the detected SMCs (see also Figure 4). Regarding the canonical terpene synthases, these occur in two major candidate families: homologues of lanosterol synthase (yeast ERG7, *n* = 33) and terpene synthase Type II homologues (*n* = 4).

The multiple copies of *carRA* homologs, emerging most likely from the ancestral rounds of whole-genome duplication [51], underpin two large groupings of terpene-type candidate SMC-encoded proteins (MCL0005/MCL0008-carRA_1, carRA_2). This conjoining of carotenoid cyclase with phytoene desaturase homologues, observed across 43 genomes (46 SMCs), suggests that the emergence of a functional carotenoid biosynthetic pathway in fungi, colocalising jointly expressed carRA+carB, could predate the *Mucoromycota*–*Dikarya* split [52]. Interestingly, an atypical hydrolase/carbonic anhydrase superfamily coding gene was identified in a typical carRA+carB carotenoid SMC (33 genomes, 36 SMCs). Apart from the possible desaturase activity, one might speculate the involvement of such an enzyme in pH buffering or in cadmium resistance. Fungi have a documented ability to deal with high cadmium concentrations [53,54] and are known to alter the pH of the substrate while growing [55].

### 3.9. Heterogeneity of NRPS/NRPS-Like Homologues

Functional NRPS clusters harbour an adenylation (A) domain, a condensation (C) domain and a thioesterase or peptidyl carrier protein (T/PCP) [56]. Due to their similarity with PKS, as well as the frequent presence of PKS/NRPS hybrid core genes, there are often SMCs bearing the features of both NRPS and PKS clusters. In Tabima et al. (2020) [39], the abundance of NRPS/NRPS-like clusters among *Mucorales* and *Mortierellales* was shown, while PKS dominated the landscape of *Zoopagales* genomes. In our study, the abundance of aryl-forming reductase (NPS) was detected in the genomes of *Mucorales*, *Endogonales* and *Diversisporales*.

In the setting of NRPS and NRPS-like clusters, MCL-based clustering revealed the limitations of rule-based SMC detection: the NRPS/NRPS-like labels could easily overlap, as the sequences are frequently fragmented (due to, for example, incomplete gene models). The largest grouping of NRPS-like sequences (MCL0003, *n* = 116, with an additional 29 SMCs labelled as NRPS; see Appendix A) concerns Class I adenylate reductases, most likely orthologous to *Schizosaccharomyces pombe* LYS2, which is an essential aminoadipate reductase involved in the primary metabolic pathway of lysine biosynthesis across fungi [57]. This observation is further corroborated by the AdenylPred results, which classified most of the MCL0003 sequences as “bulky mainly phenyl derivatives aa”, a category which encompasses lysine in the predictor output.

The moderate overlap in predictions between NRPS and NRPS-like predicted types resulted from the detection of a condensation domain in the candidate SMCs (e.g., due to matching of the conserved orthologue group signature SMCOG11207 or the fatty acid ligase similarity mentioned earlier).

An extended analysis of the putative LYS2 orthologues confirms that these candidate SMCs occur predominantly in a single copy per genome (*n* = 130; see Figure 4), with apparent multiple copies present in the assemblies of seven isolates representing the *Rhizopus*, *Mucor* and *Apophysomyces* genera. Since out of a limited number of isolates lacking the LYS2 signature (*n* = 11), the majority concern the well-characterised species of *Rhizopus oryzae* (*n* = 6) and *R. delemar* (*n* = 2); these are more likely to arise from the incompleteness of the assemblies for these particular isolates. Taken together, the ubiquity, uniqueness and readily detectable sequence signature of candidate LYS2 clusters enabled us to conclude that the conserved involvement of Type I reductases in lysine biosynthesis is likely an ancestral conserved trait across *Mucoromycota*.

Of the other subtypes of NRPS-like sequences, Type II/III adenylate-forming reductases [57] (MCL0001) were present across 127 assemblies, with the highest numbers of SMCs (total *n* = 304) in *Rhizopus microsporus* (*n* = 14), *Gigaspora margarita* (*n* = 10) and *Gigaspora rosea* (*n* = 8). The densest representation of NRPS-like SMCs was confined to members of *Mucorales* (*n* = 251) and two representatives of *Diversisporales*. Since no examples of Type IV adenylate reductases were noted (no ferredoxin-NADP+ reductase FNR domain in the *C*-terminal position [57]), one can conclude that most likely the terminal domain swap leading to a division between Type IV (L-serine reductases) and Type III (aryl acid reductases) was most likely confined to *Dikarya*. Further characterisation, including phylogenetic reconstruction on improved gene models, is necessary to unambiguously resolve whether the analysed representatives belong to Clade II (L-tyrosine reductases; indicative of a *Dikarya* specific domain swap) or III (indicative of domain swapped Type IV ancestral genes being lost in *Mucoromycota*). However, a cursory BLASTP search and an approximate maximum likelihood reconstruction of the phylogenetic tree of adenylate-forming reductases with FastTree [34] strongly suggested the latter possibility (a monophyletic clade of Type III reductases).

Prediction of substrate specificity on adenylate forming reductases has shown ‘cinnamate and succinylbenzoate derivatives’ to be the most frequent predicted class (*n* = 133 candidate SMCs). This AdenylPred class of adenylate-forming reductases encompasses enzymes involved in the production of coumarate, cinnamate, vanillate, caffeate and ferulate. The next most abundant groups of compounds were ‘luciferin’ (*n* = 57), fatty acids (*n* = 32; ‘c13 through c17′), and salicylate, 2,3-dihydroxybenzoate, anthranilate, benzoate, naphthoate xanthenurate and quinolate (*n* = 30; ‘aryl and biaryl derivatives’). Furthermore, 10 SMCs had adenylate-forming reductases predicted to utilise short organic acids (‘c2 through c5′) and aromatic amino acids and lysine (‘bulky mainly phenyl derivatives’). Taken together, the AdenylPred results suggest the substantial diversity of reductase substrates across different aromatic organic compounds of importance.

Among adenylation enzymes, we found that the *Glomeromycotina* are unique in coding for candidate SMCs containing fatty acid acyl-CoA AMP ligase (FAAL). It has been documented previously that their symbiotic relationship with plants includes the exchange not only of carbohydrates but also of lipids [58]. Pertinently, the *Glomeromycotina* are unable to produce fatty acids independently of the host plant. Apparently, they have lost Type I fatty acid synthase and retain only a mitochondrial Type II FAS synthetase of unknown molecular function. 

In stark contrast to NRPS-like adenylating enzymes, canonical nonribosomal peptide synthases were densely represented only in *Mortierella alpina* (a number ranging from 19 to 25 across three representatives) and *Gigaspora* (*G. margarita*, *n* = 13; *G.rosea*, *n* = 12).

On basis of sequence similarity (core peptide synthase), we were able to unambiguously assign putative SMCs (GCA_001021685__13_glim, GCA_000240685__19_glim, GCA_000507065__21_glim) for the recently described malpicyclin biosynthesis [12], which was present across all three *Mortierella alpina* isolates (ATCC 32222, B6842, CCTCC M207067) (Figure 5). Interestingly, this core SMC from *M. alpina* ATCC 32222 and B6842 encompasses three putative accessory genes: a saccharide deacetylase (similar to known chitin deacetylases from SwissProt), a major facilitator superfamily transporter (MFS, sugar transporter homologue) and alpha-methylacyl-CoA racemase/epimerase (also matching carnitine dehydratase SMCOG in AntiSmash predictions). While, at first glance, only the racemase accessory gene is present in the *M. alpina* CCTCC M207067 isolate, the 5′-terminal location of the SMC (relative to contig ends) suggests full biosynthetic capacity and ties with biosynthesis/transformation/efflux of glycoconjugates across *M. alpina*. AdenylPred prediction of small hydrophobic amino acid adenylation specificity fits well with the main substrate of leucine/valine residues incorporated into the cyclic peptides. The SMC is often accompanied by a zinc finger transcription factor-coding gene from the GATA zinc finger family (PF00320). The SMC is accompanied by two P-type ATPase genes and a putative toxin comprising a methylase and PIN-fold nuclease DUF1308 [59]. Proteins with such an architecture are numerous in *Glomeromycotina* (e.g., *Rhizophagus irregularis* A0A2I1GIJ1_9GLOM) and *Zoopagomycota* (*Thamnocephalis sphaerospora* A0A4P9XIN5_9FUNG) genomes and have not been characterised so far.

An additional candidate malpibaldin cluster (GCA_000507065__29_glim, *M. alpina* B6842) was selected based on sequence similarity to the MalB malpibaldin synthetase protein sequence. However, the number of gaps suggested assembly or detection issues in this case, and no putative accessory genes were annotated. The neighbouring genes are rich in protein repeat domains (Ankirin, leucine-rich repeats) and disordered regions.

The remaining NRPS SMCs identified in *Mortierellomycotina* genomes likely produce distinct cyclic peptides not characterised so far. These *Mortierellomycotina* SMCs with potentially novel compounds are likely related to the recently described malpibaldins and malpicyclins. The reductase substrates predicted by AdenylPred for all NPRS clusters are depicted in Appendix A.

### 3.10. Non-NRPS Siderophore Synthases Are Conserved in Early-Diverging Fungi

Fungal siderophores are ferric ion-specific chelators usually secreted under iron-stressed conditions and they are supposed to play crucial roles in several biological processes, including virulence or oxidative stress tolerance. The majority of fungi produce hydroxamate and carboxylate types of siderophores [60]. They can be synthesised in a nonribosomal peptide synthetase (NRPS)-dependent pathway and another NRPS-independent pathway (NRPS-independent siderophore, NIS) [61] and the first one seems to be more common and better studied (e.g., [62]). The presence of NIS enzymes was predicted in *Basidiomycota*, *Mucorales*, *Ascomycota* and *Kickxellales* representatives [61]. However, in our study, the non-NRPS siderophore synthases were shown to be conserved in early-diverging fungi.

Siderophore SMCs are characterised by a single gene coding an enzyme with a conserved protein architecture with an *N*-terminally localized IucA_IucC (PF04183) domain followed by a FhuF (PF06276) domain. The identified clusters are evolutionarily related and an approximate maximum likelihood reconstruction of the siderophore evolutionary history performed with FastTree largely recovered the species tree of fungi (Appendix A). Interestingly, we did not observe SMCs with the IucA_IucC hallmark using AntiSmash in any of *Umbelopsis* strains analysed, leading to the hypothesis of siderophore loss in this lineage. However, a BLASTP search with *M. alpina* CCTCC-M207067 GCA_001021685__6_glim-encoded siderophore synthase against the NR database led to the identification of siderophore synthetases in two recently sequenced *Umbelopsis* isolates [63] (*U. isabellina* protein acc. KAG2182392.1, *U. vinacea* KAG2188916.1), which were not included in this dataset. Siderophore SMCs seem to be vertically inherited and are present in every single clade of the early-diverging fungi sequenced so far, including Holomycota representatives such as *Fonticula alba* protein XP_009497925.1.

## 4. Conclusions

Our analysis underscores the blurry boundaries between primary and secondary metabolism, especially in the light of automated annotation based on the incomplete annotation of available resources.

On one hand, the domain/rule-based approach clearly requires supervision in order to filter out the most promising SM biosynthetic genes. As evidenced by the entanglement of NRPS-like candidate clusters with lysine biosynthesis genes, this is often insufficient to understand the relationships between distantly related SM biosynthetic gene candidates.

On the other hand, by bringing in additional, largely unsupervised lines of evidence (such as adenylation domain substrate prediction, homology to sequences of known function and intercluster similarity), the annotation can be improved considerably.

In our case, the combined approach was able to support previous findings that some aspects of secondary metabolism in *Mucoromycota* are clearly limited in taxonomic distribution (including the identified canonical NRPS clusters in *Mortierellomycotina* for the recently discovered malpibaldin and malpicyclin synthetases [12]). As evidenced by recent papers, a likely evolutionary source for divergent SMCs is the horizontal transfers from bacterial donors (the PKS-NRPS hybrid clusters found in *Diversisporales*), possibly facilitated by prior symbiosis.

However, the overview of our results also suggests that the native secondary metabolism-related toolkit of basal fungal lineages is far from scarce. Terpenoid biosynthesis, with a focus on the formation of carotenoid SM-related clusters, can be viewed as an early evolutionary trait, one shared by both *Dikarya* and *Mucoromycotina*. Retention of non-NRPS siderophore synthesis is likewise common, even in *Glomeromycotina*. There is a high diversity of adenylate forming reductases, suggesting that while clusters might not form, the early-diverging fungi do make use of an extensive array of activated substrates, which can be modified by other enzymes encoded in separate genomic loci. It remains an open question how the SMCs are regulated and when they are expressed. In the future, the desired follow-up to our work will proceed through expression studies, followed by precise delineation of cluster boundaries that can be achieved once coexpression is established. Ultimately, for the most promising candidates, this may culminate in direct characterisation of (novel) natural products.

## Figures and Tables

**Figure 1 jof-07-00285-f001:**
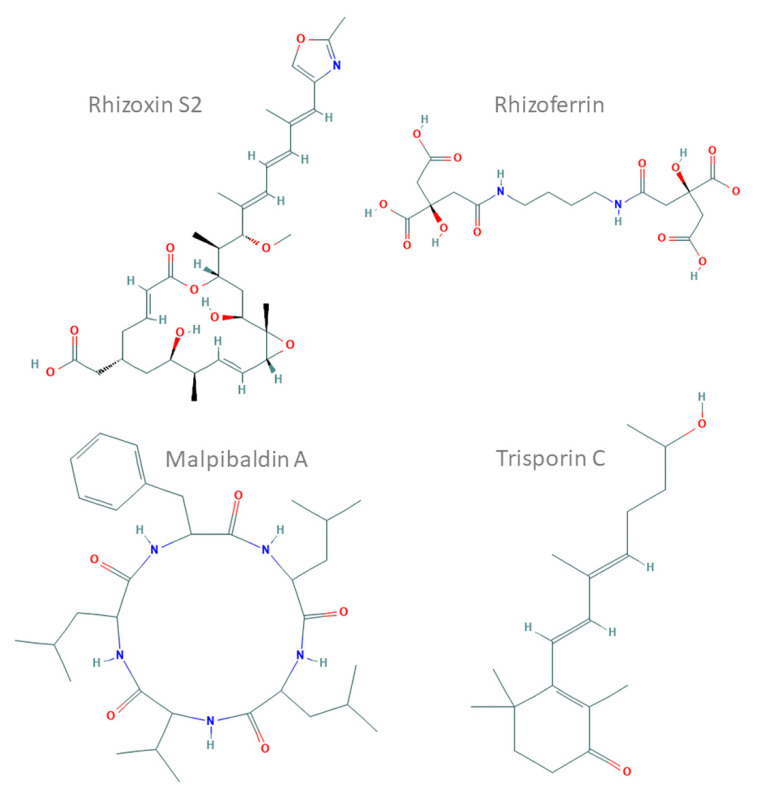
Chemical structures of representative compounds characterised in *Mucoromycotina* prepared and based on ChEMBL records (compound IDs: 44521071, 9845871, 102170144 and 72981718). Rhizoxin S2, a hybrid nonribosomal peptide synthetase (NRPS)–polyketide synthase (PKS) toxin from *Rhizopus microsporus*; rhizoferrin, a non-NRPS siderophore from *Rhizopus* spp.; malpibaldin A, an NRPS cyclopeptide from *Mortierella alpina*; trisporin C, a carotenoid from *Mucor mucedo*.

**Figure 2 jof-07-00285-f002:**
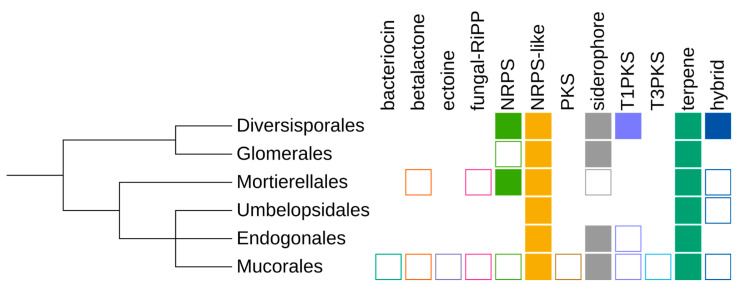
Schematic summary of the taxonomic distribution of particular secondary metabolite cluster (SMC) types. Filled rectangles, clusters of this type are present in more than 50% of isolates of the taxon; empty shape, in less than 50% of isolates; no shape, no clusters.

**Figure 3 jof-07-00285-f003:**
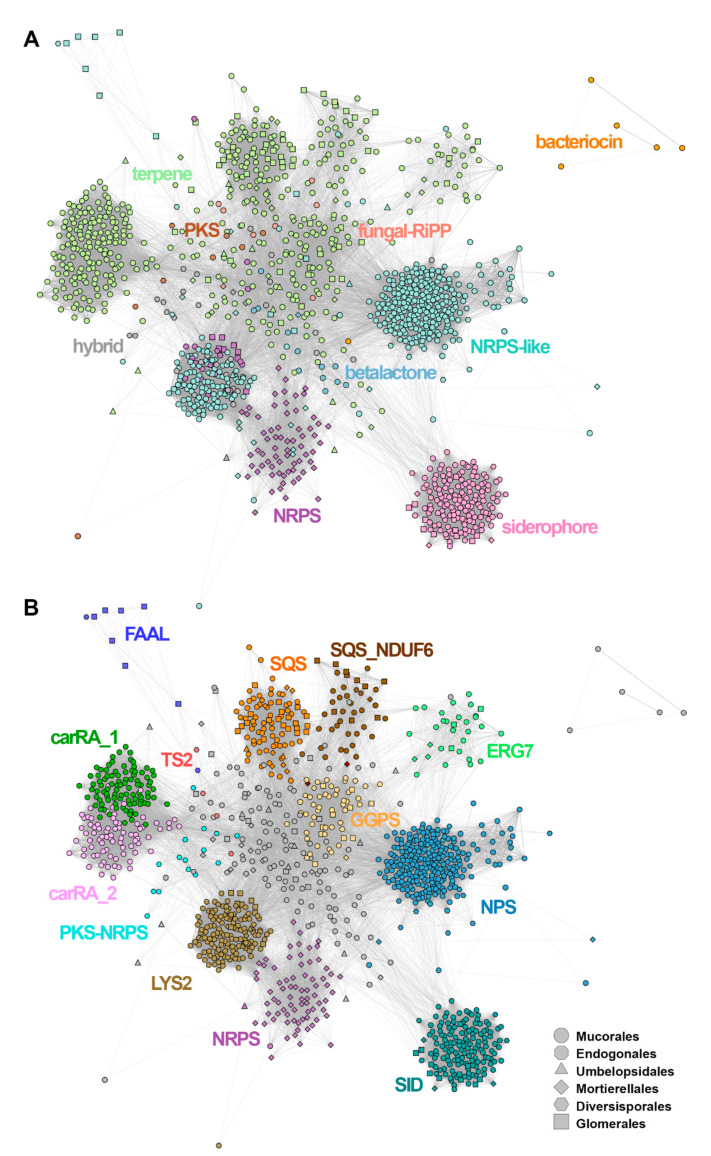
Cytoscape/MultiGeneBlast clustering of the identified SMCs, coloured according to (**A**) the predicted cluster type (AntiSmash classification rules) and (**B**) gene families: LYS2-a, aminoadipate reductases; SID, siderophore synthases (nonribosomal); SQS, squalene synthase; SQS_NDUF6, squalene synthase (homologous to NDUF6); GGPS, geranylogeranyl phosphate synthetase; ERG7, terpene synthase (homologous to lanosterol synthase); TS2, terpene synthase type 2, carRA_1, carRA_2-bifunctional lycopene cyclases/phytoene desaturases; NPS, aryl-forming reductase (nonribosomal peptide synthase-like) NRPS’, nonribosomal peptide synthase; PKS, polyketide synthase; PKS-NRPS, hybrid polyketide/nonribosomal peptide synthase; FAAL, fatty acid acyl-CoA AMP ligase.

**Figure 4 jof-07-00285-f004:**
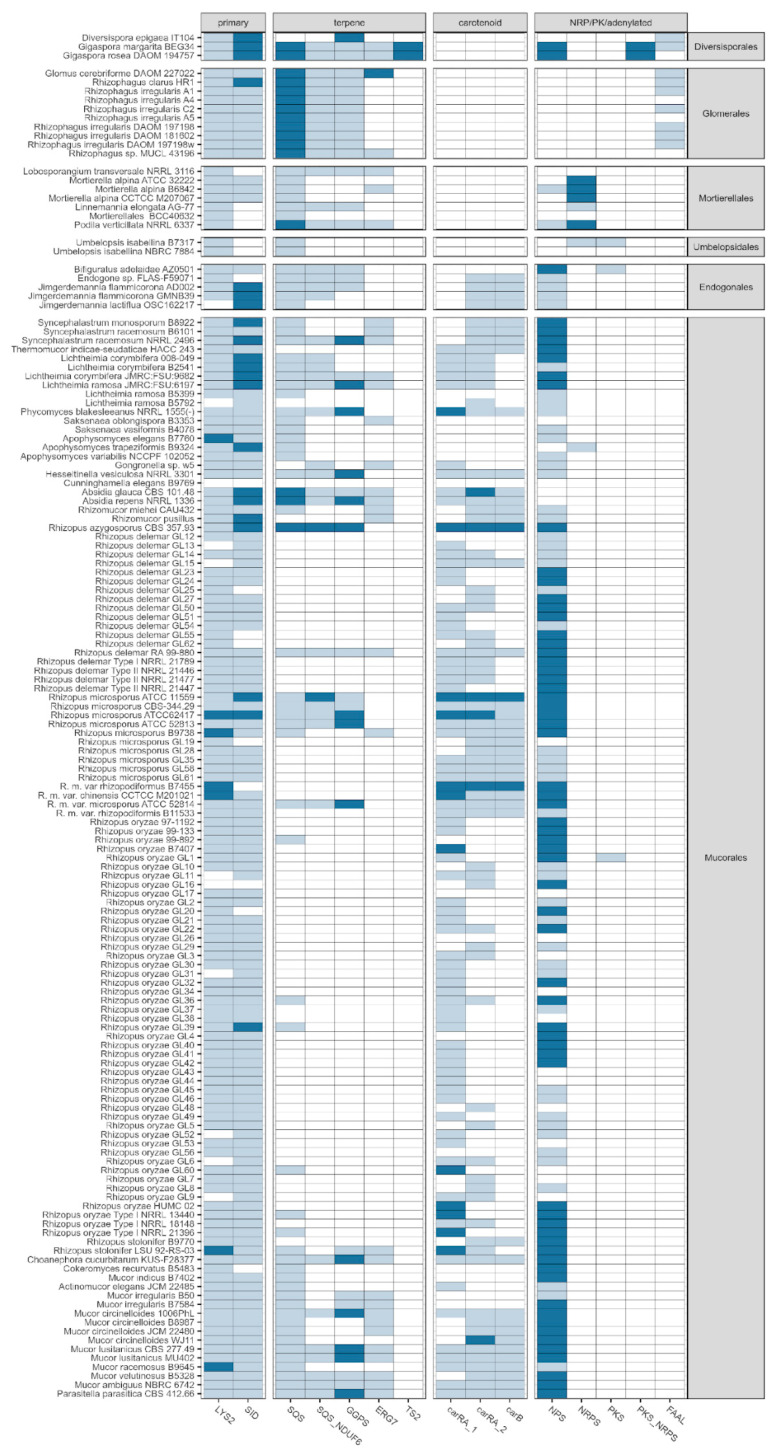
Graphical overview of the presence of crucial SMC member genes according to the results of candidate gene family annotation. Nine isolates lacking any detected SMCs, as well as GFF annotation, were left out (seven *Glomeromycota*, two *Cunninghamella bertholletae*). Gene families: LYS2-a, aminoadipate reductases; SID, siderophore synthases (nonribosomal); SQS, squalene synthase; SQS_NDUF6, squalene synthase (homologous to NDUF6); GGPS, geranylogeranyl phosphate synthetase; ERG7, terpene synthase (homologous to lanosterol synthase); TS2, terpene synthase Type 2 (annotated on the basis of the presence of the terpene cyclase-like 2 InterPro signature; IPR034686); carRA_1, carRA_2-bifunctional lycopene cyclases/phytoene desaturases; NPS, aryl-forming reductase (nonribosomal peptide synthase-like), NRPS, nonribosomal peptide synthase; PKS, polyketide synthase; PKS-NRPS, hybrid polyketide/nonribosomal peptide synthase; FAAL, (long chain) fatty acid acyl-CoA AMP ligase. Colours: white, absent; pale blue, single copy; dark blue, multiple copies per genome.

**Figure 5 jof-07-00285-f005:**
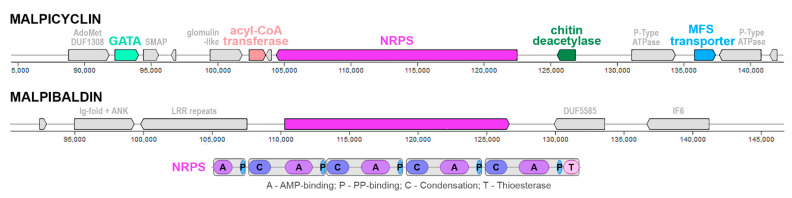
Schematic architecture of malpicyclin (GCA_000507065__21_glim) and malpibaldin (GCA_000507065__29_glim) clusters in the *M. alpina* B6842 assembly GCA_000507065. The malpibaldin NRPS region is surrounded by repeat-rich and disordered protein coding genes.

## Data Availability

All relevant data are available in the Appendix A.

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
