# Peer review of "Terpenoid Biosynthesis Dominates among Secondary Metabolite Clusters in Mucoromycotina Genomes"

_jof, 2021, doi:10.3390/jof7040285_

Round 1
Reviewer 1 Report
Manuscript describes computational analysis of secondary metabolite gene clusters (SMC) in about 50 genomes from early-diverging fungi. They are using AntiSmash program for predicting SMC, but as they are also mention in paper, output of AntiSmash greatly depends from quality of predicted genes, because in contrast to bacteria, eukaryotic (fungal) genes usually have more complex exon-intron structure,and using only ab-initio Glimmerhmm would be inadequate for for genomes where no good annotations are available. So perhaps they should restrict their analysis only to genomes, where good annotations (based on rnaseq and homology information) is available.
Besides, while AntiSmash may correctly predict backbone (or core) genes, like NRPS, PKSs or terpene synthases the prediction of correct gene clusters (multiple genes included in a cluster) is usually inaccurate for fungal genomes. So at least for some genomes with available rnaseq data, they should check if expression data collaborates with AntiSmash predictions (coexpression of genes in SMC).
Also it would good if they use also predictions from another program - SMURF (http://smurf.jcvi.org/index.php) , which in some previous papers was shown to produce
more accurate predictions of fungal PKS (PKS-like) and NRPS (NRPS-like clusters than AntiSmash
Author Response
Response to Reviewer 1 Comments
Manuscript describes computational analysis of secondary metabolite gene clusters (SMC) in about 50 genomes from early-diverging fungi. They are using AntiSmash program for predicting SMC, but as they are also mention in paper, output of AntiSmash greatly depends from quality of predicted genes, because in contrast to bacteria, eukaryotic (fungal) genes usually have more complex exon-intron structure, and using only ab-initio Glimmerhmm would be inadequate for for genomes where no good annotations are available. So perhaps they should restrict their analysis only to genomes, where good annotations (based on rnaseq and homology information) is available.
We agree that the best predictions are for genomes with annotations, however we are convinced that even fragmented predictions are of value and extend the current knowledge, as the study is the broadest so far (150 Mucoromycota assemblies analysed). Already, in the original version of the paper, we took steps to (a) clearly delimit which results are based on annotated genomes and which stem from ab initio prediction (b) discuss the limits of the approaches used. Therefore, we underline the merits of an unrestricted approach (sensitivity), where individual cases can be easily inspected based on new evidence and reanalysed in a critical follow up.
Besides, while AntiSmash may correctly predict backbone (or core) genes, like NRPS, PKSs or terpene synthases the prediction of correct gene clusters (multiple genes included in a cluster) is usually inaccurate for fungal genomes. So at least for some genomes with available rnaseq data, they should check if expression data collaborates with AntiSmash predictions (coexpression of genes in SMC).
We hope that the presented results will inspire further experimental efforts including expression analyses. Indeed, the work was conceived to facilitate in-depth analysis and validation of detected clusters (both in silico and in vitro). We fully agree that direct proof of gene activity, would be RNA-sequencing and, preferably, subsequent isolation and characterisation of metabolites. However, as it stands the RNaseq availability for Mucoromycota is rather scarce and does not cover many alternate culture/co-culture conditions. Previous studies performed in Ascomycota clearly underscore that many clusters are expressed only under specific conditions and in most media/temperature ranges optimised for laboratory cultivation remain silent (Pfannenstiel, Keller, 2019). Obtaining robust expression of a secondary cluster can be very challenging and was out of scope of in silico characterisation in the current project.
Likewise, assessing exact borders of more than one thousand candidate clusters is beyond the scope of the current project. To our best assessment, we believe it is better to err on the side of laxity in such a case - thus we maximised sensitivity by using extended ClusterFinder settings, and (where available) used predefined gene models in conjunction with ab initio annotation.
To emphasize the reviewer’s points, which we fully agree designate the limits of our approach, we added additional comments to the conclusions, strongly encouraging further experimental verification as an essential follow up.
Also it would good if they use also predictions from another program - SMURF (http://smurf.jcvi.org/index.php) , which in some previous papers was shown to produce more accurate predictions of fungal PKS (PKS-like) and NRPS (NRPS-like clusters than AntiSmash
Our results are in agreement with previous studies performed using both approaches (Antismash and Smurf) eg. Voigt and Shelest 2014, 2016 and most recent ones performed by Tabima and coworkers (2020). For classic SMCs widespread in Ascomycota, we identified only a handful of PKS clusters that is in full agreement with the aforementioned studies. These are likely not there. Likewise, NRPS clusters are also limited to one EDF lineage (Mortierellomycotina) and limited in the number of occurrences.
In our analysis, the prevailing categories for candidate SMCswere terpene and NRPS-like clusters which are rather fragmented and we do not expect better performance in SMURF. The fragmented nature of putative SMCs is also compounded due to the assembly quality and gene model issues, which would not be overcome by SMURF. In general, we agree that using more tools leads to better sensitivity of predictions. However, in this case, we would rather adhere to a single set of definitions/rules implemented in AntiSmash for clarity. The latter software has an open code which we have inspected during the experiments to check how decisions are made during the candidate SMC calling. Indeed, this has enabled us to tune the parameters during SMC prediction with and without ClusterFinder or using Glimmer v. GFF annotations. This can also be easily adjusted in the future, should evidence guide us or other researchers to either more strict or more relaxed parameterisation. We would not be able to ensure such an approach with an online tool.
Reviewer 2 Report
The manuscript “Terpenoid biosynthesis dominates among secondary metabolite clusters in Mucoromycotina genomes” by Koczyk et al. describes the genetic distribution of secondary metabolite gene clusters (SMC) in early diverging fungi, that have recently discovered to be prolific producers of natural compounds. The authors identified terpene synthases and NRPS as the major SM enzymes in basal fungi, but also consider further SM classes such as NRPS-like, NPS, NIS and PKS in their investigations. The authors combine both genetic and phylogenetic analyses with current knowledge about the biosynthetic routes of SM from Dikarya resulting in a very complex, but comprehensive overview of the metabolic potential of early diverging fungi (EDF). EDF have scarcely been analyzed by natural product researchers yet since the dogma became manifest that EDF do not produce secondary metabolites. However, the more genomes of EDF are sequenced, the more this perspective is changing. Hence, the current manuscript will support future work in various fields of life sciences including mycology, evolutionary biology, natural product research and pharmaceutical biochemistry.
The authors had to face two main issues: First, some genomes of early diverging fungi are partially fragmented and are hence incomplete, leading to a biased view on their genetic potential in production of SM. Secondly, there are only rare cases known, in which a metabolite-to-gene correlation is evident in EDF. Hence, in most cases, a biosynthetic proof of functional enzymes is lacking. Using various alternative annotation and gene prediction models and taking the biosynthetic routes of known metabolites (such as rhizoferrins) into account, the authors succeeded to give a general overview of SMCs in EDF. Nevertheless, there are some minor concerns that should be addressed:
- Because EDF are a paraphyletic group, it would be of interest to know the potential origin of their SMCs. Are all of them originated from bacterial versions (as described for Basidiobolus and Mortierella) or are there hints for an independent evolution? Are fungal SMC closely related within the EDF? Are they related to ascomycete SMC?
- In chapter 3.6 the authors discuss the bacterial origin of SMC in EDF. Putative contamination of fungal DNA with DNA of associated (endo-)bacteria is a widely observed phenomena during genome analysis of EDF. Recent investigations on Mortierella SMC revealed that they SMC genes are interspersed by introns (suggesting a eukaryotic (=fungal) gene). Are introns detectable in the SMC genes from other EDF species? Or are there bacterial regulatory elements such as the Pribnow boxes which would suggest bacterial genes?
- Are there SMCs that are unique for EDFs? It would be supportive for the community of natural product researchers to exemplarily illustrate the cluster architecture of some of the 26 novel SMCs.
- As the authors stated, some uploaded genomes of EDF are unfortunately fragmented/incomplete, i.e. they contain sequence gaps resulting in hundreds or thousands of contigs resulting in fragmented/ truncated gene clusters. Furthermore, mycelium of EDF contain multiple, genetically related but not identical nuclei, which might bias the detection of duplicate (allele-like) versions of SMC genes in the genomes. Do these two issues impact the estimation of the total number of SMC per species?
Minor remarks
Introduction: To improve the current understanding of SM biosynthesis of early diverging fungi, some representative examples of SM (chemical structures) isolated from EDF should be pictured.
Lines 56-70: To avoid confusion to readers that are not familiar with fungal siderophore biosynthesis, it should be briefly stated that siderophores are not a biosynthetic class of SM of its own, but are biosynthesized by an NRPS in case of hydroxamate-based siderophores (preferably in ascomycetes (e.g. TAFC) and basidiomycetes (e.g. basidioferrin)) or by an NIS enzyme in case of polycarboxylate siderophores (preferably in early-diverging fungi such as Mucoromycota).
Line 116: Please specify “additional supervised checks“.
Line 459: “ATC 3222” must read “ATCC 32222”
Figure legends 2 and 3: Please define “terpene synthase type 2”. Is it a diterpene synthase? What is the difference between “SQS- squalene synthase” and “SQS_NDUF6 - squalene synthase (homologous to NDUF6)”? SQS require FPP as precursor molecules: Are FPP synthases encoded in the genomes encoding SQS?
Figure 3: Strictly spoken the SM group “carotenoids” are tetraterpenoids and should be grouped in the “terpene” group.
There is some redundancy in the reference list (e.g 12 versus 62).
Inconsistency in nomenclature: Supporting Information uses “NRPS-like”, but in the main text "NPS = amino acid reductases". Please clearly specify the difference between NRPS-like and NPS.
Author Response
Response to Reviewer 2 Comments
The manuscript “Terpenoid biosynthesis dominates among secondary metabolite clusters in Mucoromycotina genomes” by Koczyk et al. describes the genetic distribution of secondary metabolite gene clusters (SMC) in early diverging fungi, that have recently discovered to be prolific producers of natural compounds. The authors identified terpene synthases and NRPS as the major SM enzymes in basal fungi, but also consider further SM classes such as NRPS-like, NPS, NIS and PKS in their investigations. The authors combine both genetic and phylogenetic analyses with current knowledge about the biosynthetic routes of SM from Dikarya resulting in a very complex, but comprehensive overview of the metabolic potential of early diverging fungi (EDF). EDF have scarcely been analyzed by natural product researchers yet since the dogma became manifest that EDF do not produce secondary metabolites. However, the more genomes of EDF are sequenced, the more this perspective is changing. Hence, the current manuscript will support future work in various fields of life sciences including mycology, evolutionary biology, natural product research and pharmaceutical biochemistry.
We thank the Reviewer for this kind opinion.
The authors had to face two main issues: First, some genomes of early diverging fungi are partially fragmented and are hence incomplete, leading to a biased view on their genetic potential in production of SM. Secondly, there are only rare cases known, in which a metabolite-to-gene correlation is evident in EDF. Hence, in most cases, a biosynthetic proof of functional enzymes is lacking. Using various alternative annotation and gene prediction models and taking the biosynthetic routes of known metabolites (such as rhizoferrins) into account, the authors succeeded to give a general overview of SMCs in EDF. Nevertheless, there are some minor concerns that should be addressed:
Because EDF are a paraphyletic group, it would be of interest to know the potential origin of their SMCs. Are all of them originated from bacterial versions (as described for Basidiobolus and Mortierella) or are there hints for an independent evolution? Are fungal SMC closely related within the EDF? Are they related to ascomycete SMC?
Since performing phylogenetic analyses for more than one thousand candidate SMCs is not feasible in the review timeframe, we explored the evolutionary scenarios in an approximate approach using the Alien Index (Gladyshev et al. 2008, Rancurel et al. 2017) configured solely to discriminate between fungal group of interest (Dikarya) and bacterial outgroup, disregarding other taxa. We corroborated evidence of transfer for NRPS, PKS/NRPS and PKS genes, particularly evident in the value of AI for designated core genes. Weaker evidence was observed for FAAL and TS2 (see below for discussion of term) clusters, and most terpene biosynthetic genes (including carotenoid) as well as adenylating reductases (NPS) were found to be a fungal development. We found that 109 out of candidate SMC were probably of prokaryotic origin, based on AI values for candidate core genes, with 89 clusters sharing transfer evidence across all predicted SMC member genes. Results of AI calculation summarised as a plot over different classes of clusters/core gene families were added as a Supplementary Figure to the materials accompanying the paper, accompanying updates were made in method, results and discussion parts. Values of index for each cluster, averaged for both the entire cluster and just the core gene families were included as additional columns the Supplementary Table as a SMC cluster attribute in ST3.
In chapter 3.6 the authors discuss the bacterial origin of SMC in EDF. Putative contamination of fungal DNA with DNA of associated (endo-)bacteria is a widely observed phenomena during genome analysis of EDF. Recent investigations on Mortierella SMC revealed that they SMC genes are interspersed by introns (suggesting a eukaryotic (=fungal) gene). Are introns detectable in the SMC genes from other EDF species? Or are there bacterial regulatory elements such as the Pribnow boxes which would suggest bacterial genes?
The two widely discussed examples of HGT (Basidiobolus and Mortierella) seem to be broadly tied to exceptional ecological niches. Analysis of intron/exon structure would require precise, prior gene calling for most of the assemblies since they lack gene structure predictions. Sadly, â…” of our dataset were genomes without initial annotation and the remaining â…“ do not necessarily have curated gene models with fully reliable intron/exon borders. In consequence, if we decided to use the current data in this way, we would likely mispredict HGT vs contamination events based on data quality issues. As data consistency was important for us, we opted to perform the same pipeline for all SMCs regardless of the genome of origin. Furthermore, we have now expanded this to include the Alien Index calculation in order to have a common framework hinting at bacterial or (shared) fungal origins in cases of different classes of SMCs and divergent taxa (whether from endosymbiont presence or ancient HGT).
Are there SMCs that are unique for EDFs? It would be supportive for the community of natural product researchers to exemplarily illustrate the cluster architecture of some of the 26 novel SMCs.
Data novelty can be assessed by different metrics, we think that many of the candidate SMCs can be novel because there were no identical clusters in MiBiG in general (based on a MultiGeneBlast search vs MiBIG).
We provide a range of characteristic features for each of the candidate SMCs in the Supplementary Table 3 and we suggest that core/accessory families listed for each cluster can serve as an approximate judgement for likely genes of importance within the broad boundaries of the clusters. Additionally, in case of NRPS/NRPS-like clusters results of AdenylPred predictions may provide some insight into the likely products.
As the authors stated, some uploaded genomes of EDF are unfortunately fragmented/incomplete, i.e. they contain sequence gaps resulting in hundreds or thousands of contigs resulting in fragmented/ truncated gene clusters. Furthermore, mycelium of EDF contain multiple, genetically related but not identical nuclei, which might bias the detection of duplicate (allele-like) versions of SMC genes in the genomes. Do these two issues impact the estimation of the total number of SMC per species?
In general, assessment of completeness/duplication of SMCs repertoire for assemblies of divergent quality is hard, and assembly quality measures such as BUSCO are useless for generally rare objects. We found that gene calling quality matters a lot in SMCs identification (which is summarized in the Supplementary Tables). We focused then on the presence of SMC sequences instead of the absence. We did not observe many occurrences of highly similar/identical SMCs in one assembly. However we observed almost identical SMC in assemblies from related isolates/strains/species suggesting vertical inheritance. These are particularly visible as high scores in MultiGeneBlast (Supplementary Table) and the visual representation in Cytoscape (see chapter 3.2 Limits of de novo prediction in absence of reference annotation).
Minor remarks
Introduction: To improve the current understanding of SM biosynthesis of early diverging fungi, some representative examples of SM (chemical structures) isolated from EDF should be pictured.
There is a very limited number of published and deposited chemical structures of SMs isolated from EDF. We prepared a schematic representation of four compounds rhizoferrin (Rhizopus), rhizoxin S2 (Rhizopus microsporus), malpibaldin A (Mortierella alpina ) and trisporin C (Mucor mucedo) and have included this in the main text of the paper.
Lines 56-70: To avoid confusion to readers that are not familiar with fungal siderophore biosynthesis, it should be briefly stated that siderophores are not a biosynthetic class of SM of its own, but are biosynthesized by an NRPS in case of hydroxamate-based siderophores (preferably in ascomycetes (e.g. TAFC) and basidiomycetes (e.g. basidioferrin)) or by an NIS enzyme in case of polycarboxylate siderophores (preferably in early-diverging fungi such as Mucoromycota).
This has been corrected in text and underlined in the caption to Figure 1.
Line 116: Please specify “additional supervised checks“.
This has been clarified in the text. By checks, we meant (a) inspection of MiBIG results to identify core/accessory genes (b) assignment of descriptive names to clusters based on their protein domain composition and similarity.
Line 459: “ATC 3222” must read “ATCC 32222”
This has been corrected.
Figure legends 2 and 3: Please define “terpene synthase type 2”. Is it a diterpene synthase? What is the difference between “SQS- squalene synthase” and “SQS_NDUF6 - squalene synthase (homologous to NDUF6)”? SQS require FPP as precursor molecules: Are FPP synthases encoded in the genomes encoding SQS?
Along the manuscript, we use assigned names originating from underlying protein family databases and/or similarities to UniProt/SwissProt proteins. In the case of TS2, the description originated from detected PFAM domain name Terpene_syn_C_2 (PF19086) and InterPro Terpene cyclase-like 2 (IPR034686). In case of SQS/SQS_NDUF6, the former denotes the canonical squalene synthase found in fungal SMCs and the latter has similarity to NDUFAF6 (NDUF6, mitochondrial complex assembly factor 6) which is known to occur across the tree of life and was previously found to contain a squalene/phytoene synthase domain, albeit with changes in DxxD motif suggesting abolished catalysis, hence the abbreviated distinction SQS_NDUF6. The two proteins form separable groups in Markov clustering. However, we did not study in detail the relationship between squalene synthase families or functionality of EDF homologs of SQS/SQS_NDUF6. That clarification has been added to the manuscript.
As for FPP synthases, these are not treated as a cluster core genes in AntiSmash framework, nevertheless we conducted an impromptu TBLASTN search to verify its presence, using Penicillium brevicompactum FDPS as a query. As expected, the gene is ubiquitous. Only in 3 cases FDPS signature was not present in the assembly (2 where neither SQS nor FDPS are present, 1 when SQS was detected but no FDPS signature). Therefore, we can safely conclude that farnesyl pyrophosphate synthesis is indeed both retained as a crucial prerequisite of further biosynthesis and detectable at current levels of assembly completeness.
Figure 3: Strictly spoken the SM group “carotenoids” are tetraterpenoids and should be grouped in the “terpene” group.
We decided to delimit this group, as it is particularly well studied and characterised. We clarified in the text, that carotenoids are part of the terpenoid category of compounds, delineated as a subgroup for ease of discussion and somewhat unique position among the best known compounds.
There is some redundancy in the reference list (e.g 12 versus 62).
That has been fixed.
Inconsistency in nomenclature: Supporting Information uses “NRPS-like”, but in the main text "NPS = amino acid reductases". Please clearly specify the difference between NRPS-like and NPS.
NRPS-like is a denomination used by AntiSmash for type of clusters based on the presence of adenylating domain without other requisite parts of full non-ribosomal peptide synthase. NPS is used as a shorthand for adenylate forming reductases based on majority of these genes, first tentatively described in Basidiomycetes, having a moniker starting with NPS/Nps (e.g. Nps1 from Cerisporiopsis subvermispora - Kalb et al. 2014). We added an explanation of the abbreviation as NPS - non- (nonribosomal)peptide synthase=adenylate forming reductase to the material and methods section, paragraph 2.2, where the term is used for the first time.
Round 2
Reviewer 1 Report
Authors provide some arguments why their paper is still publishable and I in principle agree, that although study is based mostly on computational predictions it is still has some value for subsequent experimental follow up of predictions